# Large gains in schooling and income are possible from minimizing adverse birth outcomes in 121 low- and middle-income countries: A modelling study

Mia M. Blakstad[1]*, Nandita Perumal[1], Lilia Bliznashka[1], Mark J. Lambiris[2,3], Günther Fink[2,3], Goodarz Danaei[1,4], Christopher R. Sudfeld[1,5]

1 Department of Global Health and Population, Harvard TH Chan School of Public Health, Boston, MA, United States of America, 2 Department of Epidemiology and Public Health, Swiss Tropical and Public Health Institute, Basel, Switzerland, 3 University of Basel, Basel, Switzerland, 4 Department of Epidemiology, Harvard T.H. Chan School of Public Health, Boston, MA, United States of America, 5 Department of Nutrition, Harvard TH Chan School of Public Health, Boston, MA, United States of America

* mib273@hsph.harvard.edu

**Data Availability Statement:** All data used is publicly available and accessible through the sources referenced in the manuscript (see

## Abstract

While the global contributions of adverse birth outcomes to child morbidity and mortality is relatively well documented, the potential long-term schooling and economic consequences of adverse birth outcomes has not been estimated. We sought to quantify the potential schooling and lifetime income gains associated with reducing the excess prevalence of adverse birth outcomes in 121 low- and middle-income countries. We used a linear deterministic model to estimate the potential gains in schooling and lifetime income that may be achieved by attaining theoretical minimum prevalence of low birthweight, preterm birth and small-for-gestational age births at the national, regional, and global levels. We estimated that potential total gains across the 121 countries from reducing low birthweight to the theoretical minimum were 20.3 million school years (95% CI: 6.0,34.8) and US$ 68.8 billion (95% CI: 20.3,117.9) in lifetime income gains per birth cohort. As for preterm birth, we estimated gains of 9.8 million school years (95% CI: 1.5,18.4) and US$ 41.9 billion (95% CI: 6.1,80.9) in lifetime income. The potential gains from small-for-gestational age were 39.5 million (95% CI: 19.1,60.3) school years and US$113.6 billion (95% CI: 55.5,174.2) in lifetime income gained. In summary, reducing the excess prevalence of low birthweight, preterm birth or small-for-gestational age births in low- and middle-income countries may lead to substantial long-term human capital gains in addition to benefits on child mortality, growth, and development as well as on risk of non-communicable diseases in adults and other consequences across the life course.

## Introduction

Globally, it is estimated that 14.6% of all live births are low birthweight (LBW, birthweight <2500g), 10.6% are preterm (PTB, <37 weeks of gestation), and 27.0% are small-for-

Supplement 1). The prevalences of adverse birth outcomes were obtained from Lee et al 2013, Chawanpaiboon 2019, and Blencowe 2019.1-3 Birth cohorts and survival probabilities were obtained from the United Nations Population Division World Population Prospects 2019 (https://population.un.org/wpp/). Data on wages were obtained from the World Indicators Database (https://data.worldbank.org/indicator/NY.GDP.MKTP.KD). Survival probabilities were obtained from Institute for Health Metrics Life tables (https://www.healthdata.org) and the United Nations Population Division World Population Prospects 2019 (https://population.un.org/wpp/).

**Funding:** This study and its authors were funded by the Bill and Melinda Gates Foundation (OPP1198520). NP was partially supported by the Canadian Institutes of Health Research Fellowship. The funders had no role in study design, data collection and analysis, decision to publish, or preparation of the manuscript.

**Competing interests:** The authors have declared that no competing interests exist.

gestational age (SGA, birth weight below the 10th percentile of gestational age- and sex relative to the standard reference population) [1–3]. While LBW, PTB, and SGA can co-occur in the same child the population-level prevalence of these adverse birth outcomes by country and setting [4]. Complications from adverse birth outcomes are major causes of death in children under 5 years of age with preterm birth as the current leading cause of global child mortality [1–4]. In addition, prematurity and SGA (an indicator of intrauterine growth restriction) are associated with increased risk of child linear growth faltering and suboptimal development [5, 6]. LBW and PTB may also have long-term consequences including an increase in the risk of non-communicable diseases in adulthood, including heart disease and diabetes [7–9].

While the consequences of adverse birth outcomes on child health, nutrition, and development are well-documented, evidence on the potential downstream effects on human capital consequences are much more limited [7]. To the best of our knowledge, only one study has estimated the potential economic benefits of eliminating LBW in LMICs. This study estimated a gain of $US 510 per infant that moved from LBW to non-LBW status due to reductions in costs associated with infant and child mortality, illness and care, stunting, suboptimal cognitive ability, and increased risk of chronic disease [10]. However, this study assumed the same 15% prevalence of LBW across all LMICs and did not estimate economic benefits of reducing other adverse birth outcomes such as PTB and SGA. In addition, this prior study modeled the scenario of reducing LBW prevalence to 0% globally which is biologically implausible given that the prevalence of LBW was 3.2% among births in the INTERGROWTH-21st fetal growth reference study [11]. In this modeling paper, we quantified the potential gains in school years and lifetime income that may be gained from reducing the current prevalence of LBW, SGA, and PTB in LMICs to the theoretical minimum prevalence.

## Methods

We used a population attributable risk framework to estimate the potential human capital gains in school years and lifetime income for reducing the excess prevalence of LBW, SGA, PTB from their 2015 national prevalence in 121 LMICs to the theoretical minimum prevalence levels [12]. We used a framework previously conceptualized to estimate the potential human capital benefits of scaling up prenatal nutrition interventions on schooling and lifetime income [12] In this study we applied a theoretical approach to estimate the potential benefits of reducing the prevalence of adverse birth outcomes to the minimum prevalence observed in a population of healthy pregnancies.

### Counterfactual exposure distribution

Risk assessment requires a reference scenario, a hypothetical counterfactual comparison group. The theoretical minimum-risk exposure distribution (TMRED) is a distribution of exposures corresponding to the lowest levels of risk. In the case of this analysis, the TMRED represents the theoretical prevalence of adverse birth outcomes in a population of healthy pregnancies in low- and middle-income countries. We used evidence from the INTERGROWTH-21st population-based study to define the theoretical minimal prevalence of LBW, SGA and PTB [11]. From a cohort of 59,137 women across eight countries, the project selected 20,486 women meeting strict individual eligibility criteria for pregnancies with low risk of impaired fetal growth to construct normative anthropometric standards for newborn size for use across multiethnic populations. In the absence of socio-economic, health, or nutritional risk factors, LBW prevalence was 3.2% and PTB prevalence was 5.5%. In addition, by definition, the theoretical prevalence of SGA (defined by the 10th percentile) in a population of healthy pregnancies is 10%. Notably, these adverse birth outcomes are not mutually exclusive

(i.e., infants born LBW and/or PTB can be SGA or appropriate-for-gestational age [13]) and because we lacked country-level estimates of their joint distributions, we estimated the potential human capital impacts of LBW, PTB, and SGA births independently. For countries with prevalences at or below the theoretical minimum, the expected gains in school years and lifetime income were set to 0. Therefore, for each country, we estimated the absolute percentage point difference between current excess prevalence of each adverse birth outcome and the theoretical optimal exposure risk threshold.

## Assessment of exposure distribution

This analysis uses secondary datasets that are publicly available through the referenced sources. We included 121 LMICs as defined by the World Bank July 2019 income classification [14]. We obtained the most recent country-specific prevalence estimates for LBW, PTB, and SGA from global modeling analyses in 2015 (S1 Table) [1–3]. We used 2010 prevalence estimates for LBW when newer estimates were not available [3]. For countries for which prevalence of birth outcomes were unavailable, we performed random-effects meta-analyses to impute the sub-regional level averages and variances, as defined by the Global Burden of Disease (GBD) regions [15]. Where 2010 LBW prevalences were reported without associated variance, we used the largest variance in the GBD subregion from the 2015 data [1]. Eleven Eastern European countries were excluded from the analysis because SGA prevalences were missing and no other GBD subregion prevalence estimates were available. We also excluded South Sudan because SGA prevalence estimates were not available and there was too high variation in SGA estimates of the neighboring counties within the GBD subregion to confidently impute the prevalence [3].

## Effect of adverse birth outcomes on schooling

We conducted a *de novo* systematic review and meta-analysis of the link between birthweight and human capital outcomes later in life [16]. The majority of studies identified in the systematic review were from high-income countries; as such, we used random-effects meta-analysis to derive a pooled estimate of the effect of LBW on attained schooling by combining evidence from the *de novo* systematic review and data from five birth cohorts in LMICs [7, 17]. It was estimated that on average LBW-born individuals complete -0.29 (95% CI: -0.48, -0.10) fewer school years (Table 1) [12]. For PTB and SGA, estimates were available from the COHORTS collaboration [7]. PTB-born individuals were estimated to complete an average of -0.32 (95% CI: -0.57, -0.06) fewer school years, while SGA-born individuals complete an average of -0.41 (95% CI: -0.62, -0.20) fewer school years (Table 1).

**Table 1. Effect size sources for model parameterization.**

| Data input | Value (Mean difference ± 95% CI) | Source | Method |
|---|---|---|---|
| Estimated effect on birth outcome on schooling attainment | | | |
| LBW | -0.2894 years (-0.483, -0.096) | Adair et al. 2013 | Pooled estimate combining evidence from *de novo* review of the economics literature and Adair et al. estimates |
| PTB | -0.317 years (-0.572, -0.062) | Adair et al. 2013 | Unpublished estimates from Adair et al. from COHORTS |
| SGA | -0.41 years (-0.62, -0.20) | Stein et al. 2013 | From COHORTS data |
| Estimates for returns on wage gains (%) per additional year of schooling | Varies by country. Examples: Bangladesh 7.1%, Burkina Faso 6.3%. | Fink et al., 2016 | Fink et al. for LMICs outside Europe. Peet et al. for LMICs in Europe. |
| | | Peet et al. 2015 | |

We multiplied the effect sizes linking birth outcomes to educational attainment with the absolute percentage point change in each adverse birth outcome to obtain the average school years gained per child due to reductions in LBW, PTB, or SGA prevalence independently. We then used the school years gained per child, multiplied by each country's 2015 birth cohort size and the country-specific probability that the child would survive until age 25, to quantify the estimated total gains in years of schooling completed [18].

### Effect of education on net present value of future income

We quantified the gains in adult earning potential using effect estimates of returns in income for each added year of schooling for each country [19]. To calculate the net present value of future lifetime income we summed annual income between ages 20 and 59 for each country. In line with previous estimations, annual incomes were estimated at $2/3^{rds}$ of gross domestic product as reported by the World Development Indicators [20, 21]. We used 2010 constant US $, corrected for a 3% discounting rate, a 2% real income growth rate, and the country-specific survival probabilities for each working year based on the sex-average survival probabilities estimated by the Institute for Health Metrics and Evaluation [22]. The total gains in lifetime income per birth cohort was estimated by multiplying the gains in income per child by the country-specific birth cohort size. We further estimated the percent increase in net present value of lifetime income per child by dividing the increase in per child lifetime income by the net present value of working from 20 to 59 years. Five countries (Dominica, Republic of Kosovo, Marshall Islands, Nauru, and Tuvalu) were excluded from the analysis because they lacked data on the number of live births and probability of survival.

A working example for our methods available in the S1 Text.

### Uncertainty estimation

We used a first order Monte Carlo simulation with 1,000 iterations to propagate uncertainty of each parameter of the model. Uncertainty estimates were available from published estimates for LBW, PTB, and SGA prevalences, for the effect sizes of the association between adverse birth outcomes and years of schooling, and between adverse birth outcomes and returns on income per additional year of education. All model parameters were assumed to be normally distributed and to be independent as they are derived from different studies. The 95% confidence intervals were drawn from the $2.5^{th}$ and $95.5^{th}$ percentile of from the distribution of 1,000 iterations. All analyses were performed using Stata 16.1 Statistical Software package (StataCorp LP).

## Results

Across the 121 LMICs included in the analysis, median prevalences of LBW, PTB, and SGA were 11.5% (IQR: 8.2,15.4), 10.4% (9.8–12.0), and 20.4% (13.6–25.1), respectively (Table 2).

The estimated schooling and income gains attributed to excess LBW, PTB, and SGA is presented by region in Table 2 with country-specific estimates in S1 Table. When each country's prevalence of LBW was reduced to the theoretical minimum of 3.2%, the estimated gains in educational attainment were 20.3 million (95% CI 5.98, 34.8) school years and US$ 68.8 billion (95% CI: 20.3,117.9) in net present value of lifetime income across all 121 LMICs. The largest absolute gains in schooling years were estimated in South Asia (10.7 million; 95% CI: 3.1,19.4) and sub-Saharan Africa (5.4 million; 95% CI: 1.6,9.2). India had the largest total potential gains from reductions in LBW with 7.8 million (95% CI: 2.1,15.0) school years and US$ 19.8 billion (95% CI: 5.2, 38.0) per birth cohort. In addition, Fig 1 present the estimated increases in educational attainment and lifetime earnings per child (scaled by population): The

**Table 2. Gains in years of schooling and net present value of lifetime income gains.**

| Birth outcome | Birth cohort size[1] | Low birthweight (<2500g) | | | Preterm birth (<37 weeks' gestation) | | | Small for gestational age (<10th percentile) | | |
|---|---|---|---|---|---|---|---|---|---|---|
| | | Prevalence in 2015 Median, IQR[2] | Years of Schooling[3] (mln) | Income gains[4] (US$ bln) | Prevalence Median, IQR | Years of Schooling (mln) | Income gains (US$ bln) | Prevalence Median, IQR | Years of Schooling (mln) | Income gains (US$ bln) |
| Central Asia (n = 9) | 10.0 | 5.5 (5.3–6.7) | 0.67 (0.01,0.12) | 0.3 (0.1,0.6) | 10.4 (10.2–10.4) | 0.12 (0.02,0.23) | 0.3 (0.0,0.7) | 15.4 (12.9–16.5) | 0.18 (0.07,0.33) | 1.0 (0.3,2.1) |
| Latin America and Caribbean (n = 24) | 50.6 | 9.7 (7.9–11.1) | 0.82 (0.25,1.43) | 14.1 (4.3,24.5) | 9.8 (9.0–9.8) | 0.63 (0.09,1.19) | 12.0 (1.7,23.9) | 13.4 (11.3–16.0) | 0.57 (0.25,1.06) | 8.7 (2.6,19.4) |
| North Africa and Middle East (n = 14) | 57.9 | 9.3 (6.9–14.7) | 1.15 (0.34,2.00) | 6.6 (1.9,11.5) | 10.4 (10.3–13.4) | 1.01 (0.15,2.21) | 5.8 (0.9,13.2) | 13.4 (9.8–19.7) | 1.94 (0.94,3.09) | 9.7 (4.1,18.1) |
| South Asia (n = 5) | 168.8 | 26.0 (16.8–27.7) | 10.74 (3.09,19.42) | 24.1 (6.9,44.1) | 10.4 (6.8–16.4) | 3.66 (0.55,7.37) | 8.3 (1.3,16.7) | 39.6 (30.5–47.0) | 23.03 (11.18,35.85) | 52.4 (25.5,81.7) |
| Sub-Saharan Africa (n = 47) | 188.7 | 14.4 (12.0–16.9) | 5.35 (1.59,9.21) | 13.9 (4.2,24.5) | 12.0 (12.0–12.0) | 3.06 (0.46,5.92) | 8.3 (1.2,17.0) | 24.6 (21.9–29.0) | 10.27 (4.90,15.98) | 25.4 (12.0,38.5) |
| Southeast Asia, East Asia, and Oceania (n = 22) | 146.9 | 11.0 (8.2–12.4) | 1.92 (0.60,3.35) | 8.7 (2.5,16.3) | 10.0 (10.0–10.4) | 1.27 (0.21,2.49) | 6.3 (1.1,13.6) | 20.3 (17.5–23.9) | 3.31 (1.49,5.22) | 15.3 (6.7,27.4) |
| All LMICs (n = 121) | 622.9 | 11.5 (8.2–15.4) | 20.31 (5.98,34.78) | 68.8 (20.3,117.9) | 10.4 (9.8–12.0) | 9.80 (1.45,18.38) | 41.9 (6.1,80.9) | 20.4 (13.6,25.1) | 39.52 (19.13,60.31) | 113.6 (55.5,174.2) |

Gains in years of schooling and net present value of lifetime income gains from reducing adverse birth outcomes to the theoretical minimum prevalence. Theoretical minimum risk exposure distributions are 3.2% for low birthweight, 5.5% for preterm birth, and 10% for small for gestational age. Values in parentheses are 95% uncertainty intervals based on bootstrapped SEs. 1: Size of birth cohort in million live births by GBD super region. 2: Interquartile range. 3: Total years of schooling (in millions) gained per birth cohort. 4: Net present value of lifetime wage gains per birth cohort.

percentage increase in the net present value of lifetime income per child were largest in Sudan (0.78%, 95% CI: 0.22, 1.40), Morocco (0.62%, 95% CI: 0.18, 1.12), and the Comoros (0.58%, 95% CI: 0.18, 1.15). Potential gains were highest in Latin America and the Caribbean, with absolute gains in net present value in lifetime income per child born at US$ 279 per child born (95% CI: 84.4,85.5; Fig 2; LBW) and per-LBW child gains at US$ 3,042 (95% CI: 919, 5,284; Fig 3).

When each country's prevalence of PTB was reduced to the theoretical minimum of 5.5%, the estimated gains in schooling years across all 121 countries were 9.8 million (95% CI: 1.5, 18.4) years, with the largest gains seen in South Asia (6.7 million school years; 95% CI: 0.6,7.4). Overall gains in net present value of lifetime income were US$ 41.9 billion (95% CI: 6.1, 80.9), with South Asia (US$ 8.3 billion; 95% CI: 1.4,16.7) and sub-Saharan Africa (US$ 8.3 billion; 95% CI: 1.2,17.0) each contributing US$ 8.3 billion per birth cohort. Fig 1 shows that increases in educational attainment per 1,000 children were high across Africa and South Asia, whereas percentage increases in net present value of lifetime income were highest in the Southern Africa and South Asia regions. The three countries with the highest percentage increase in the net present value of lifetime income per child were South Africa (0.34%, 95% CI: 0.0, 0.9), Morocco (0.33%, 95% CI: 0.0, 1.2), and eSwatini (0.30%, 95% CI: 0.0,0.6). Potential gains were highest in Latin America and the Caribbean, with absolute gains in net present value in lifetime income per child born at US$ 2236 (95% CI: 33, 472; Fig 2; PTB) and per-LBW child gains at US$ 2,392 (95% CI: 339, 4,775; Fig 3).

When each country's prevalence of SGA was reduced to the theoretical minimum of 10%, the estimated gains in schooling were 39.5 million (95% CI: 19.1,60.3) school years across 121 LMICs, with the largest gains seen in South Asia (23.0 million school years; 95% CI: 11.2,35.9).

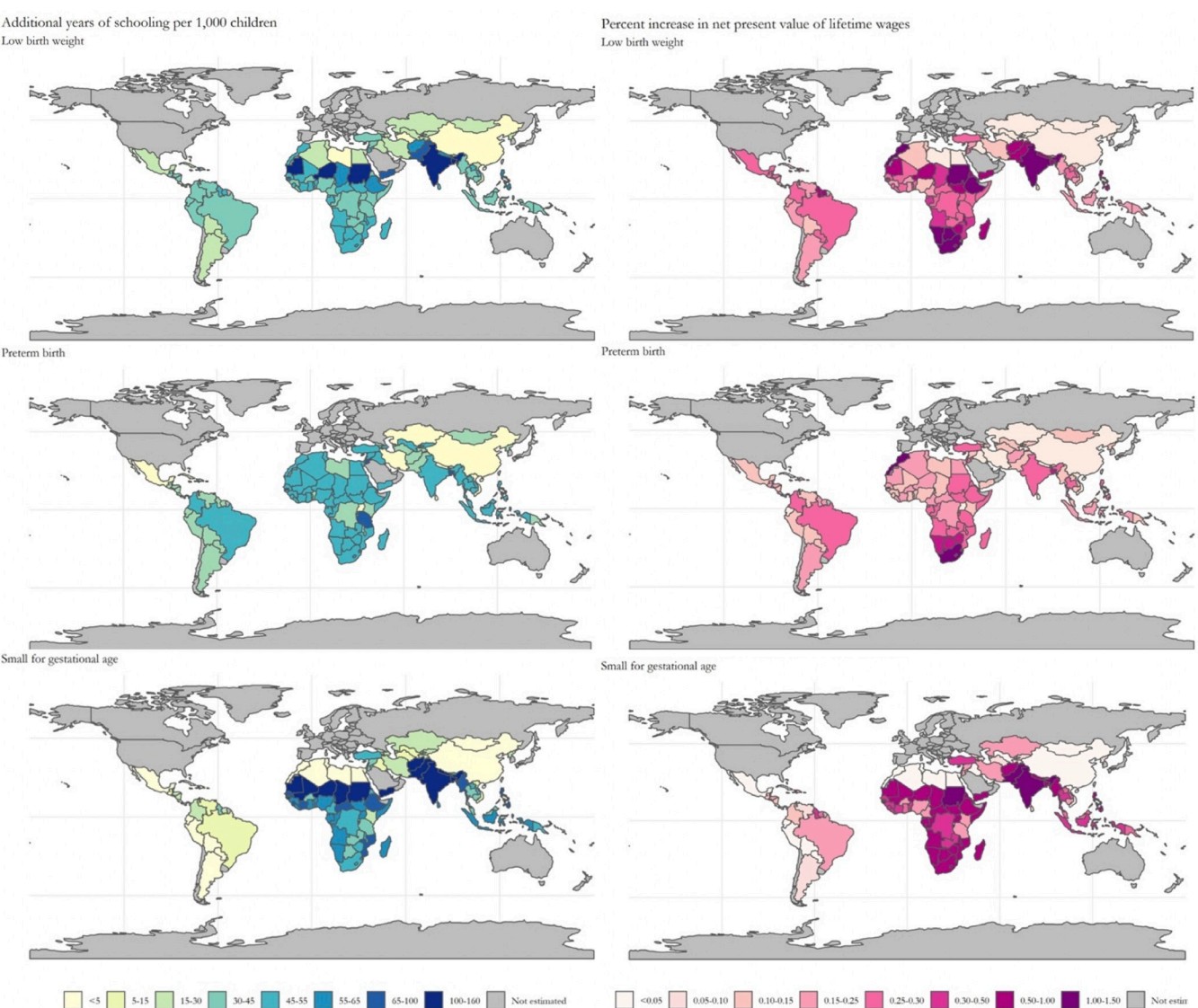

**Fig 1. Absolute gains in schooling and relative increase in annual income by country.** Maps were made using the spData package in R and basemaps from
https://www.naturalearthdata.com/.

Overall potential gains in net present value of lifetime income across all included LMICs was US$ 113.6 billion (95% CI: 55.5,174.2). Both increase in educational attainment per 1,000 children and percentage increase in net present value of lifetime income were highest in sub-Saharan Africa and South Asia; at the country-level, the percentage increase in net present value of lifetime wages was largest in Sudan (1.27%, 95% CI: 0.6,2.1), Pakistan (1.14%, 95% CI:0.5,1.8), and India (1.13%, 95% CI:0.5,1.8) (Fig 1). Absolute potential gains in net present value highest in South Asia, at US$ 311 per child born (95% CI: 151,484; Fig 2), but potential gains per child born SGA were highest in Latin America at US$ 1,252 (95% CI: 373, 2,808; Fig 3).

## Discussion

Through this analysis, we found that there may be substantial potential gains in schooling and lifetime income to gain from reducing adverse birth outcomes to theoretical minimum values

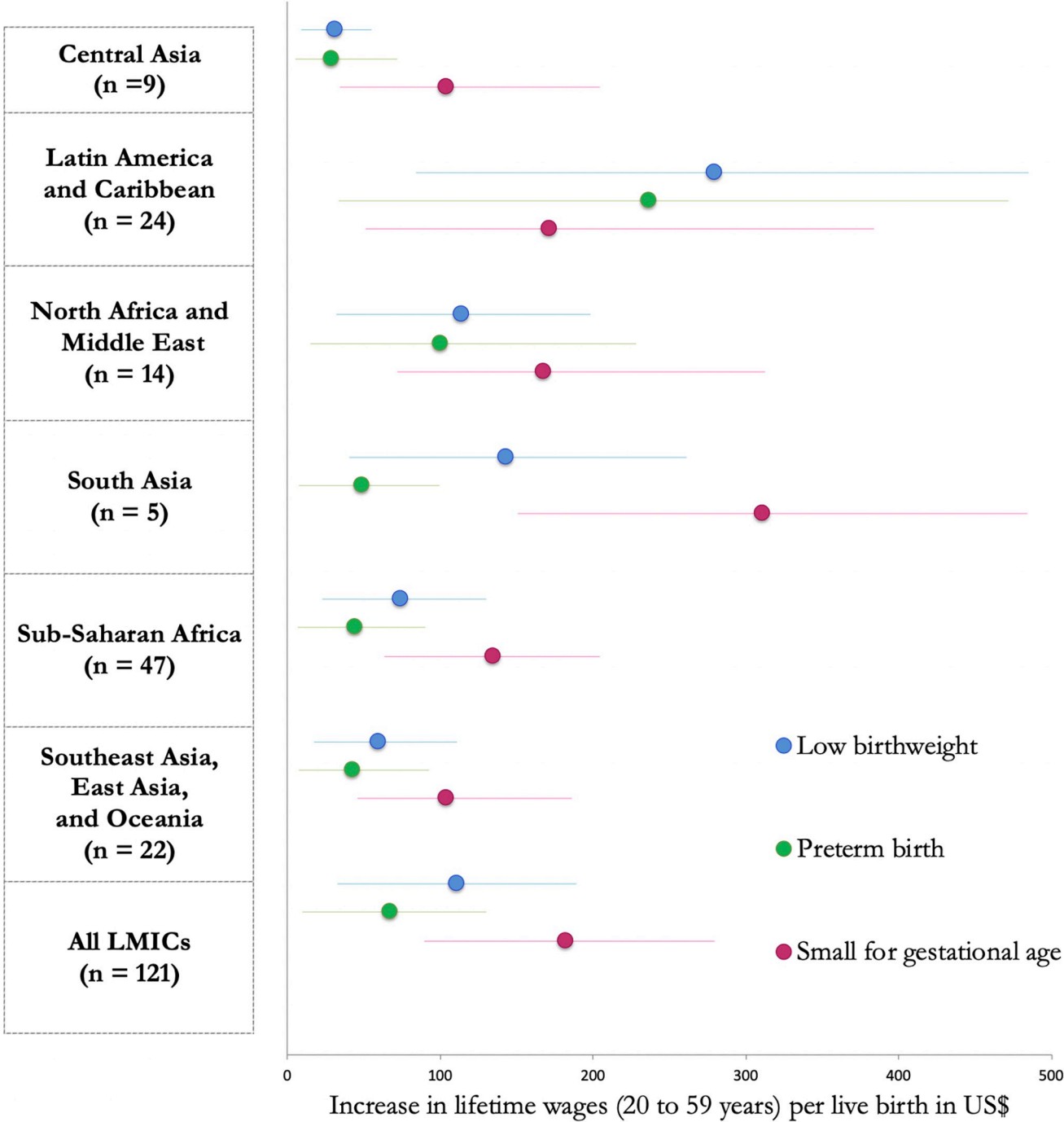

**Fig 2. Population average gains in lifetime income per child born in US$, by region.**

across 121 LMICs. The estimated total gains in additional school years and lifetime income from shifting the excess prevalence of LBW, PTB, and SGA to the theoretical minimum levels were 20.3 million, 9.8 million, and 20.3 million school years, and US$ 68.8 billion, US$ 41.9 billion, and US$113.6 billion in lifetime income gained per birth cohort, respectively. While

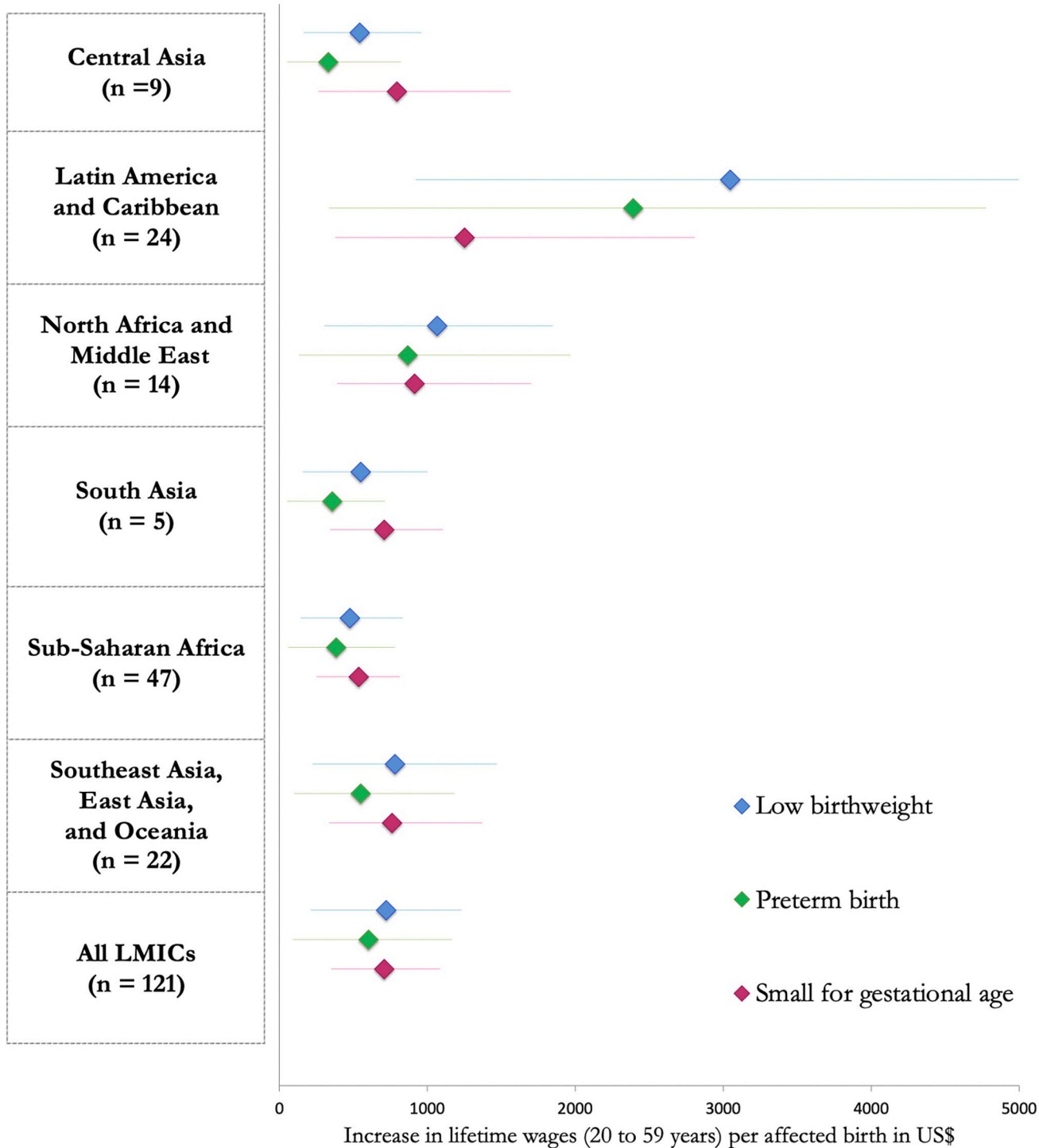

**Fig 3. Gains in lifetime income per child born low birthweight, preterm, or small-for-gestational age in US$, by region.** Maps were made using the spData package in R.

the education and income gains were estimated to be large for each adverse birth outcome considered, the potential gains were generally largest for reducing SGA to the theoretical minimum of 10%. Notably, South Asia and sub-Saharan Africa had substantial potential gains in

absolute and per-child terms for education attainment and lifetime earnings largely owing to the high prevalence of adverse birth outcomes and large population of these regions.

Our findings add to a very limited literature on the potential long-term economic returns of reducing birth outcomes and early life exposures and adversities in LMICs [10, 20]. To our knowledge, only one other study has looked at the potential economic gains of reducing the prevalence of LBW: Alderman and Behrman estimated that relative to infants born LBW, non-LBW infants may have a greater lifetime earning potential of US $832 (at 3% discounting) [10]. Alderman and Behrman used different methods and investigated a wider range of potential gains, including not only cognitive delay, but also reduced costs from infant mortality, neonatal care, sickness, stunting, and chronic diseases. Of that, US $367 were estimated as productivity gains from increased intelligence quotients and schooling attainment, in the absence of LBW [10]. In comparison, we estimated a global average potential gain of US$ 717 per LBW case averted. Our results suggest that the returns on wage gains from increased schooling may be almost twice that estimated in their analysis.

We report the first estimates the potential education and lifetime earnings gains to reducing PTB and SGA. It is important to note that the large gains in sub-Saharan Africa and South Asia can in part be attributed to the higher prevalence of PTB and SGA in these regions. There were particularly high returns to reducing SGA to the TMRED in South Asia, where the median prevalence of SGA born babies was 40%. Several studies have found significant associations between prematurity and birthweight for gestational age with cognitive performance of school-aged children [23, 24].

Our study builds on data from birth cohort studies that linked adverse birth outcomes with poorer education outcomes. In terms of mechanisms, there are multiple pathways from adverse birth outcomes to reduced number of completed years of schooling, including suboptimal cognitive, executive function, socioemotional and other neurodevelopmental pathways. There is a relatively large literature in both high-income and LMIC that have linked prematurity and LBW with poorer child neurodevelopment outcomes [5, 23–25]. Animal and human studies have also found that that in-utero malnutrition is associated with reduced brain volume and with adverse effects on neuron proliferation, synaptogenesis, and myelination [26]. In addition, higher birthweight have been associated with greater lifetime earnings in data from high-income settings [16]. However, little to no data on the relationship of adverse birth outcomes with lifetime earnings exists for LMICs and therefore we needed to use schooling as a mediator.

Our analytical strategy included analyses based on not just LBW, but also PTB and SGA, and enables a comparison of potential impacts across each type of adverse birth outcome. This was done because LBW does not differentiate between prematurity and intrauterine growth restriction and each mechanism may have different long-term effects on schooling and life income. The inclusion of PTB and SGA is an important addition to the limited literature on potential long-term human capital gains of reducing adverse birth outcomes. It is also important to note that LBW, PTB, and SGA can be coexisting conditions; therefore, the impacts of reducing each on schooling and income should be interpreted independently rather than cumulatively. We were not able to calculate the joint effect of the three outcomes due to lack of country-level estimates of their joint distribution. Nevertheless, the joint effect would be larger than each individual effect, but smaller than the sum of all three. Therefore, the impacts of each birth outcome should be interpreted independent of the other adverse birth outcomes.

Our study has several limitations and strengths. Our estimates are limited by the availability and quality of relevant data. Estimates of adverse birth outcomes estimates from nationally representative surveys are not available in many LMICs, and we therefore used the latest modeled prevalence estimates of LBW, PTB and SGA, which are limited by the quality of the data

used in the models. Further, five countries (Dominica, Republic of Kosovo, Marshall Islands, Nauru, and Tuvalu) were excluded from the analysis because they lacked data on the number of live births and probability of survival. Ultrasound dating to assess gestational age is not common in LMICs and therefore estimation of the national prevalences of PTB and SGA at the population-level is particularly challenging in LMICs. However, our study also has several strengths. We used a first order Monte Carlo simulation propagated uncertainty in LBW, PTB and SGA are were therefore able account for variation in parameters and compute bootstrapped confidence intervals [27]. In addition, we quantified the impact of reducing adverse birth outcomes to their theoretical minima on lifetime earnings exclusively through educational attainment. Therefore, our estimates for lifetime earnings are likely conservative since they do not take into account for additional benefits on lifetime earnings that may be accrued through pathways outside of schooling, such as reductions in morbidity, well-being, and the associated increases in adult productivity and lower healthcare costs [10, 28, 29]. In addition, reductions in adverse birth outcomes would reduce child mortality that would lead to more children surviving, enrolling in school, and leading productive lives which are additional benefits on top of the ones accounted for in our model.

The potential education and lifetime earnings gains from reducing LBW, PTB, and SGA globally may be substantial. The long-term gains documented in this analysis should be considered in cost-effectiveness evaluations of interventions that improve birth outcomes in addition to more immediate benefits on child mortality, growth, and development. As a result, greater investment in interventions and programs that improve birth outcomes may benefit individuals across the life course and provide substantial population-level human capital returns.

## Supporting information

**S1 Table. Data sources.**
(DOCX)

**S1 Text. Working example.**
(DOCX)

**S1 Data. Country-specific estimates.**
(DOCX)

**S1 Code. Stata code used for analysis.**
(DO)

## Author Contributions

**Conceptualization:** Nandita Perumal, Goodarz Danaei, Christopher R. Sudfeld.

**Data curation:** Mia M. Blakstad, Nandita Perumal, Lilia Bliznashka, Mark J. Lambiris.

**Formal analysis:** Mia M. Blakstad, Lilia Bliznashka, Goodarz Danaei.

**Funding acquisition:** Günther Fink, Christopher R. Sudfeld.

**Investigation:** Mia M. Blakstad, Nandita Perumal, Christopher R. Sudfeld.

**Methodology:** Mia M. Blakstad, Nandita Perumal, Mark J. Lambiris, Günther Fink, Goodarz Danaei, Christopher R. Sudfeld.

**Resources:** Mark J. Lambiris, Günther Fink.

**Supervision:** Günther Fink, Goodarz Danaei, Christopher R. Sudfeld.

**Validation:** Mia M. Blakstad.

**Visualization:** Mia M. Blakstad, Lilia Bliznashka.

**Writing – original draft:** Mia M. Blakstad.

**Writing – review & editing:** Nandita Perumal, Lilia Bliznashka, Mark J. Lambiris, Günther Fink, Goodarz Danaei, Christopher R. Sudfeld.

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
