## [Decision Letter · Decision Letter 0]

7 Dec 2021

PGPH-D-21-00493

Large gains in schooling and income are possible from minimizing adverse birth outcomes in 121 low- and middle-income countries: a modelling study

Dear Dr. Blakstad,

Thank you for submitting your manuscript to PLOS Global Public Health. After careful consideration, we feel that it has merit but does not fully meet PLOS Global Public Health’s publication criteria as it currently stands. Therefore, we invite you to submit a revised version of the manuscript that addresses the points raised during the review process.

We look forward to receiving your revised manuscript.

Kind regards,

Dickson Abanimi Amugsi, PhD

Academic Editor

Journal Requirements:

2. In your ethics statement, please provide additional information about the data used in your retrospective study. Specifically, please ensure that you have discussed whether all data were fully anonymized before you accessed them.

3. We ask that you please remove citations for unavailable and unpublished work, including manuscripts that have been submitted but not yet accepted (e.g., “unpublished work,” “data not shown”). Instead, include those data as supplementary material or deposit the data in a publicly available database.

4. Please update the completed 'Competing Interests' statement, including any COIs declared by your co-authors. If you have no competing interests to declare, please state "The authors have declared that no competing interests exist". Otherwise please declare all competing interests beginning with the statement "I have read the journal's policy and the authors of this manuscript have the following competing interests:"

5. Please ensure that all Supporting Information files are included correctly and that each one has a legend listed in the manuscript after the references list. 

6. Please amend your Data Availability Statement and indicate where the data may be found

7. Please provide us with a direct link to the base layer of the map used in Figure 1 and ensure this location is also included in the figure legend. 

Please note that, because all PLOS articles are published under a CC BY license (creativecommons.org/licenses/by/4.0/), we cannot publish proprietary maps such as Google Maps, Mapquest or other copyrighted maps. If your map was obtained from a copyrighted source please amend the figure so that the base map used is from an openly available source.

Please note that only the following CC BY licences are compatible with PLOS licence: CC BY 4.0, CC BY 2.0  and CC BY 3.0, meanwhile such licences as CC BY-ND 3.0 and others are not compatible due to additional restrictions. If you are unsure whether you can use a map or not, please do reach out and we will be able to help you. 

The following websites are good examples of where you can source open access or public domain maps:

8. Please amend your detailed Financial Disclosure statement. This is published with the article, therefore should be completed in full sentences and contain the exact wording you wish to be published.

i) Please include all sources of funding (financial or material support) for your study. List the grants (with grant number) or organizations (with url) that supported your study, including funding received from your institution. 

ii). State the initials, alongside each funding source, of each author to receive each grant.

iii). State what role the funders took in the study. If the funders had no role in your study, please state: “The funders had no role in study design, data collection and analysis, decision to publish, or preparation of the manuscript.”

Additional Editor Comments (if provided):

Thank you for submitting your paper to PGPH for publication. The reviewers felt it presents an interesting piece of work. However, they raised a number of issues (attached) that will need your attention before the paper can be published in PGBH.

Further, after going through the same, I have also identified the under-listed issues for your consideration:

1. Lines 81-84: A bit more detail explanation of the frameworks will be helpful to the reader---the paper should be self-containing

2. Lines 107, 112-118, and 151, need to be referenced

3. Take out the ORs and CIs completely from the discussion section. This section is not meant for the restatement of the results, but to discuss what the results mean, and in the context of the literature.

4. The discussion section needs to be thoroughly revised---in some places it sound more like the restatement of results than discussion

5. What the results mean and thier connection with the extant literature did not come out strongly. As an example, the connection between the findings of your study and the existing literature in lines 260-271 is not clear.

6. I was expecting to see a bit of discussion on the analytical strategies used in the context of the results (e.g. superior findings, or otherwise, as a result of employing the same)

7. Data quality issues (quality of studies etc) have not been discussed.

8. What are the strengths and limitations of the study? These should be spelt out clearly in a separate paragraph.

9. There should be a paragraph/section on the conclusions/recommendations of the study

The paper appears strong, therefore, addressing the reviewers as well as my comments will bring it to a publishable state.

Reviewers' comments:

Reviewer's Responses to Questions

**Comments to the Author**

1. Does this manuscript meet PLOS Global Public Health’s publication criteria? Is the manuscript technically sound, and do the data support the conclusions? The manuscript must describe methodologically and ethically rigorous research with conclusions that are appropriately drawn based on the data presented.

Reviewer #1: Yes

Reviewer #2: Yes

2. Has the statistical analysis been performed appropriately and rigorously?

Reviewer #1: Yes

Reviewer #2: Yes

3. Have the authors made all data underlying the findings in their manuscript fully available (please refer to the Data Availability Statement at the start of the manuscript PDF file)?

Reviewer #1: Yes

Reviewer #2: Yes

4. Is the manuscript presented in an intelligible fashion and written in standard English?

Reviewer #1: Yes

Reviewer #2: Yes

5. Review Comments to the Author

Reviewer #1: The manuscript entitled “Impact of birth outcomes on education and wages in adulthood” was organized based on PLOS Global Public Health’s publication criteria. The manuscript is grammatically well organized. The manuscript was technically sound. The study used meta-analysis; it is a good strength of the manuscript and indicated as it is a methodological manuscript. But the author must re-write the methodology part with a detailed explanation of statistical analysis methods. I have no concern about the dual publication of the manuscript. Generally it is suitable for publication.

Reviewer #2: The paper was well thought out and written.

The explanation on fixing the theoretical minimum prevalence needs additional explanation on parameters included for modelling.

The inclusion criteria for the subjects from the cohort should be given clearly.

The negative sign of the estimates in line 128 and line 130 is missing but presented in Table.1.

The mention of 95% C.I is not there in line 130 and line 131.

Recommendations:

The low and middle income countries have high prevalence of LBW, closer to 20%. Hence it will be helpful for the policy makers if the potential gain was calculated for higher prevalence of 10% also.

It will be helpful for the readers if the authors show the calculation of school years gain, Increase in life time earning and Benefits by cohort lifetime wages for one country with all needed information.

6. PLOS authors have the option to publish the peer review history of their article (what does this mean?). If published, this will include your full peer review and any attached files.

**Do you want your identity to be public for this peer review?** For information about this choice, including consent withdrawal, please see our Privacy Policy.

Reviewer #1: No

Reviewer #2: No

---

## [Editor Report · Decision Letter 1]

1 Apr 2022

Large gains in schooling and income are possible from minimizing adverse birth outcomes in 121 low- and middle-income countries: a modelling study

PGPH-D-21-00493R1

Dear Dr Blakstad,

We are pleased to inform you that your manuscript 'Large gains in schooling and income are possible from minimizing adverse birth outcomes in 121 low- and middle-income countries: a modelling study' has been provisionally accepted for publication in PLOS Global Public Health.

Best regards,

Dickson Abanimi Amugsi, PhD

Academic Editor